# Gut microbiota-CRAMP axis shapes intestinal barrier function and immune responses in dietary gluten-induced enteropathy

Zhengnan Ren[1,2,†] (iD), Li-Long Pan[3,†], Yiwen Huang[1,2], Hongbing Chen[4], Yu Liu[5], He Liu[1,2], Xing Tu[1,2], Yanyan Liu[1,2], Binbin Li[1,2], Xiaoliang Dong[1,2], Xiaohua Pan[1,2], Hanfei Li[6,7], Yu V Fu[6,7], Birgitta Agerberth[8], Julien Diana[9,*,‡] (iD) & Jia Sun[1,2,**,‡] (iD)

## Abstract

In the gut, cathelicidin-related antimicrobial peptide (CRAMP) has been largely described for its anti-infective activities. With an increasing recognition of its immune regulatory effects in extra-intestinal diseases, the role of CRAMP in gluten-induced small intestinal enteropathy celiac disease remains unknown. This study aimed to investigate the unexplored role of CRAMP in celiac disease. By applying a mouse model of gluten-induced enteropathy (GIE) recapitulating small intestinal enteropathy of celiac disease, we observed defective CRAMP production in duodenal epithelium during GIE. CRAMP-deficient mice were susceptible to the development of GIE. Exogenous CRAMP corrected gliadin-triggered epithelial dysfunction and promoted regulatory immune responses at the intestinal mucosa. Additionally, GIE-associated gut dysbiosis with enriched *Pseudomonas aeruginosa* and production of the protease LasB contributed to defective intestinal CRAMP production. These results highlight microbiota-CRAMP axis in the modulation of barrier function and immune responses in GIE. Hence, modulating CRAMP may represent a therapeutic strategy for celiac disease.

**Keywords** antimicrobial peptides; celiac disease; gluten-induced enteropathy; microbiota; interleukin-15
**Subject Categories** Digestive System; Immunology; Microbiology, Virology & Host Pathogen Interaction

## Introduction

Antimicrobial peptides (AMPs) or host defence peptides (HDPs) are evolutionarily conserved peptides as an essential innate defence mechanism found in most all living organisms (Zasloff, 2002; Gallo & Hooper, 2012). Apart from their classical antimicrobial activity, the immune modulatory capacity of AMPs has been increasingly appreciated and they may cause either pro-inflammatory or anti-inflammatory effects dependent on the disease model being investigated and their cellular source (Hancock *et al*, 2016). Among AMPs, we and others have particularly highlighted the role of cathelicidins [the sole member named LL-37 in humans and cathelicidin-related antimicrobial peptide (CRAMP) in mice] in autoimmune diseases. Commonly, neutrophil-derived CRAMP, often citrullinated and when in excess, forms complexes with self-nucleic acids to activate plasmacytoid dendritic cells (pDCs), inducing deleterious autoimmune responses (Diana *et al*, 2013; Kahlenberg & Kaplan, 2013). However, we previously demonstrated that pancreatic β-cells-derived CRAMP induces regulatory immune cells to shape pancreatic immune microenvironment, thereby protecting against autoimmune diabetes in non-obese diabetic (NOD) mice (Sun *et al*, 2015).

Not surprisingly, the mammalian gut epithelium produces a diverse collection of AMPs, responding to the complex environmental cues in the intestine. In the gut, the role of cathelicidins has been largely confirmed to the large intestine for colonic inflammatory diseases or its anti-infective activities (Mitsutoshi *et al*, 2005; Raqib *et al*, 2006; Hing *et al*, 2013; Mukherjee & Hooper, 2015). To a lesser extent, cathelicidin expression is found in the small intestine (Gallo *et al*, 1997; Mukherjee & Hooper, 2015). Despite an increasing

---

1 State Key Laboratory of Food Science and Technology, Jiangnan University, Wuxi, China
2 School of Food Science and Technology, Jiangnan University, Wuxi, China
3 Wuxi Medical School, Jiangnan University, Wuxi, China
4 State Key Laboratory of Food Science and Technology, Nanchang University, Nanchang, China
5 Department of Endocrinology and Metabolism, Sir Run Run Shaw Hospital, Nanjing Medical University, Nanjing, China
6 State Key Laboratory of Microbial Resources, Institute of Microbiology, Chinese Academy of Sciences, Beijing, China
7 Savaid Medical School, University of Chinese Academy of Science, Beijing, China
8 Department of Laboratory Medicine, Division of Clinical Microbiology, Karolinska Institutet, Karolinska University Hospital Huddinge, Stockholm, Sweden
9 Institut Necker Enfants Malades (INEM), Institut National de la Santé et de la Recherche Médicale (INSERM), Paris, France
  *Corresponding author. Tel: +33 06 85 40 67 34; E-mail: julien.diana@inserm.fr
  **Corresponding author. Tel: +86 510 85197370; E-mail: jiasun@jiangnan.edu.cn
  †These authors contributed equally to this work
  ‡These authors contributed equally to this work as senior authors

---

recognition of its role in autoimmune disease contexts, the role of cathelicidins in the development of dietary gluten-induced small intestinal enteropathy with autoimmune features such as celiac disease remains to be explored.

Here, we observe defective CRAMP production in duodenal epithelium of mice with gluten-induced enteropathy (GIE). While CRAMP-deficient ($Cnlp^{-/-}$) mice are susceptible to the development of GIE, increasing duodenal CRAMP prevents GIE, by shaping intestinal barrier function and immune response. GIE-associated dysbiosis with enhanced *Pseudomonas aeruginosa* (*P. aeruginosa*) contributes to CRAMP degradation via production of the protease LasB. Thus, the current study reveals a critical role of CRAMP in modulating GIE and supports future therapeutic strategy targeting CRAMP for the prevention of celiac disease.

# Results

## Epithelial cathelicidin production is defective in dietary gluten-induced enteropathy (GIE)

To test our hypothesis, we first determined the relevance of cathelicidin production and regulation with GIE in mice. In order to evaluate intestinal CRAMP expression and its effect on gluten-induced small intestinal enteropathy, a mouse model of GIE by adoptive transfer of gliadin-presensitized $CD4^+CD45RB^{low}CD25^-$ T cells into lymphopenic $Rag1^{-/-}$ mice, recapitulating intestinal pathology of human celiac disease (Freitag *et al*, 2009) was applied (Fig 1A). Serum CRAMP levels were found much lower in mice received gliadin-presensitized T cells and fed with gluten-containing diet (gluten group) than in those fed with gluten-free diet (gluten-free group; Fig 1B). Notably, we observed no difference in serum CRAMP levels in unsensitized mice fed with gluten-free diet or gluten-containing diet (Fig EV1A). Western blot experiment confirmed CRAMP expression in duodenum of $Rag1^{-/-}$, C57BL/6 and BALB/c mice but not of CRAMP-deficient mice under steady state (Fig EV1B). After GIE induction, CRAMP became scarcely expressed both for the pro-form (18 kDa) and the secreted mature form (5 kDa) in duodenal tissue and in *ex vivo* isolated duodenal epithelial cells (Fig 1C and D). CRAMP may be expressed by epithelial cells and by infiltrating immune cells (Gallo & Hooper, 2012). Immunofluorescent staining confirmed that CRAMP staining (red) was primarily localized in intestinal epithelium and co-expressed with E-cadherin-positive cells, instead of $Ly6G^+$ neutrophils or $F4/80^+$ macrophages which were largely present in the lamina propria (Fig 1E). CRAMP expression by epithelial cells was reduced in mice fed with gluten diet, despite of increased Ly6G and F4/80 staining indicative of increased immune infiltrates. Lastly, we examined whether an association of serum LL-37 levels, the human ortholog of CRAMP and susceptibility to clinical celiac disease could be established. Serum samples of celiac disease high-risk subjects positive for anti-tissue transglutaminase immunoglobulin A antibodies (anti-tTG IgA) or/and anti-deamidated gliadin peptide immunoglobulin G antibodies (anti-DGP IgG)] and double-negative subjects (anti-tTG $IgA^-$ anti-DGP $IgG^-$) were obtained. Intriguingly, we found lower levels of serum LL-37 in celiac disease high-risk subjects than in double-negative subjects (Fig EV1C). Together, these results suggest that CRAMP

may play a role in GIE and prompt us to investigate how CRAMP modulates GIE development.

## CRAMP deficiency potentiates GIE

As CRAMP production is defective during GIE development, we investigated the impact of CRAMP deficiency on GIE development. Classic symptoms of GIE include weight loss and duodenal histological changes (Freitag *et al*, 2009; Fig 2A). CRAMP-deficient mice with GIE ($Cnlp^{-/-}$/gluten group) exhibited worsened weight loss compared with control mice (gluten group; Fig 2B). Similarly, characteristic histological changes including duodenal crypt hyperplasia, villus atrophy and lymphocytic infiltration (Freitag *et al*, 2009) were exacerbated in CRAMP-deficient mice induced with GIE ($Cnlp^{-/-}$/gluten group; Fig 2C). Intestinal tight junction proteins (TJPs) locate at the apical ends of the lateral membranes of intestinal epithelial cells, including zona occludens-1 (ZO-1), ZO-2, occludin and claudin-1, and maintain the intestinal barrier integrity (Suzuki, 2013). Dysregulation of small intestinal TJPs leading to increased intestinal permeability is a key pathological event of GIE (Alessio, 2012). As shown in Fig 2D, loss of duodenal TJPs was exacerbated in $Cnlp^{-/-}$/gluten group, compared with gluten group. Zonulin is a unique protein so far known to reversibly regulate intestinal permeability by modulating intercellular TJP expression (Amit *et al*, 2009; Alessio, 2012). Duodenal zonulin was found highly produced in $Cnlp^{-/-}$/gluten group (Fig 2E), leading to suppressed TJP expression and increased intestinal permeability (Fig 2F). Collectively, CRAMP deficiency aggravates GIE.

## Replenishing duodenal CRAMP is protective against GIE

We further investigated the effect of exogenous CRAMP treatment on GIE development. Mice were intraperitoneally administered with CRAMP (100 μg mouse$^{-1}$ week$^{-1}$) either two weeks before (prophylactic treatment) or 6 weeks after (therapeutic treatment) adoptive gliadin-sensitized T cells transfer (Fig 3A). Small intestinal CRAMP levels were replenished following CRAMP administration (Figs 3B and EV2A and B). GIE-associated weight loss was notably attenuated by both prophylactic and therapeutic treatment with CRAMP (Fig 3C). We found that increasing duodenal CRAMP protected against gluten-induced gut barrier dysfunction as evidenced by improved histological markers (duodenum in Fig 3D, jejunum in Fig EV2C and ileum in Fig EV2D), increased expression of TJPs (duodenum in Fig 3E, jejunum in Fig EV2E and ileum in Fig EV2F), decreased duodenal zonulin production (Fig 3F) and reduced intestinal permeability (Fig 3G). Together, these data support a protective role of CRAMP for GIE by shaping intestinal barrier integrity.

## CRAMP counteracts gliadin-triggered inflammatory signalling to protect epithelial function

Next, we attempted to decipher signalling mechanism underlying the regulatory effect of CRAMP on epithelial barrier function. Gliadin, the undigested immunogenic peptide from gluten, is known to induce prolonged epidermal growth factor receptor (EGFR) activation at Tyr1068, leading to downstream phosphoinositide-3-kinase/protein kinase B (Pi3K/Akt)-nuclear factor kappa light chain

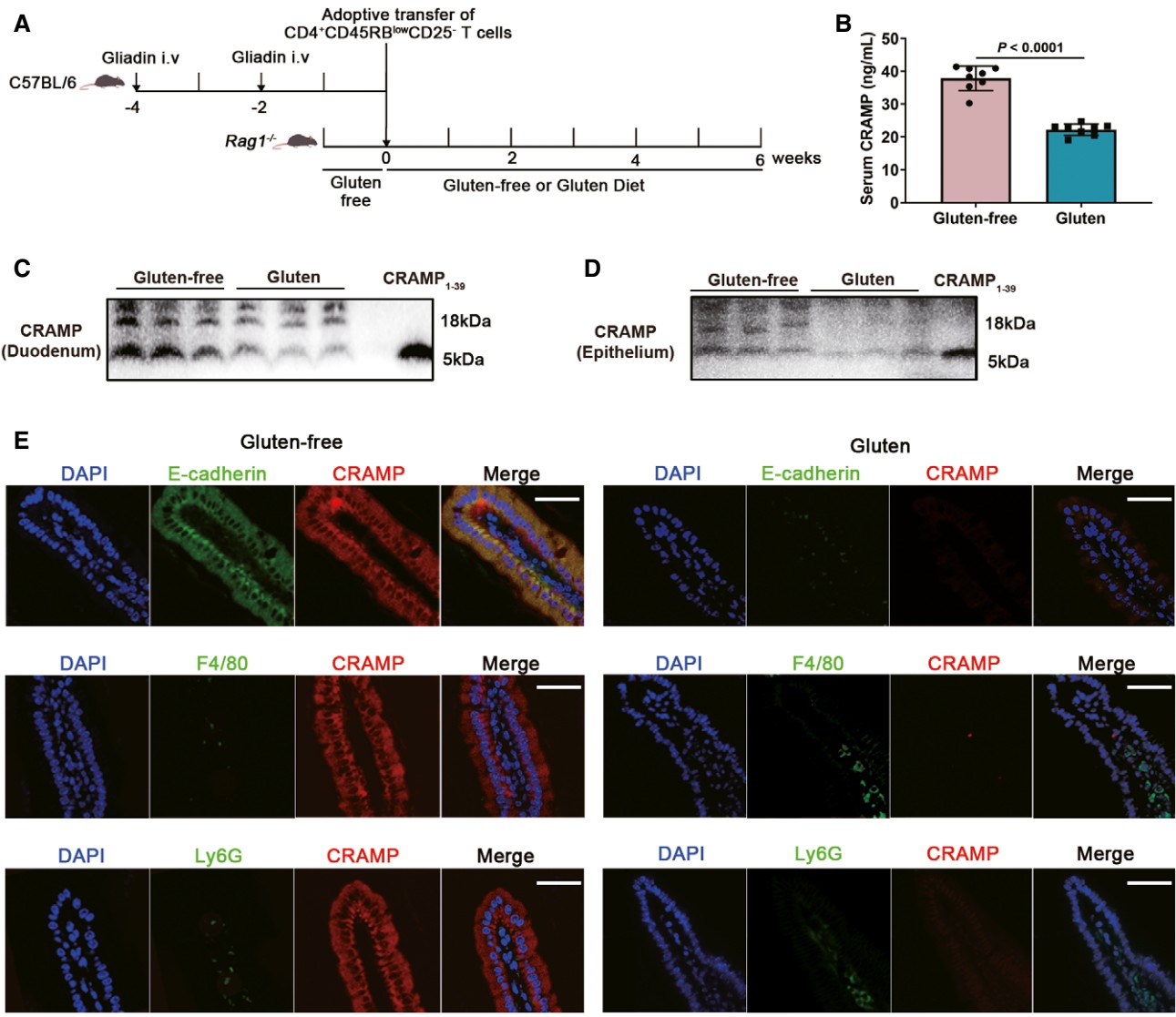

**Figure 1.  Epithelial CRAMP production is defective in dietary gluten-induced enteropathy (GIE).**

A    Animal protocol.

B    Serum CRAMP determination by ELISA ($n = 8$).

C, D  Western blot of pro-form CRAMP and mature peptide in duodenum and *ex vivo* duodenal epithelial cells.

E    Localization and expression of CRAMP (red), E-cadherin (green), F4/80 (green) and Ly6G (green) in duodenum by immunofluorescent staining. Representative photomicrographs of individual and merged staining were shown. Nuclei were stained with DAPI (blue). Scale bar: 50 μm.

Data information: Data in B were representative and were the mean ± SD from three independent experiments. Data (C–E) were representative from three independent experiments. *P* values were calculated by unpaired two-tailed *t*-test for comparison of two groups.

Source data are available online for this figure.

enhancer of activated B cells (NF-κB) signalling or a myeloid differentiation factor 88 (MyD88)-dependent zonulin production, triggering epithelial barrier disruption (Thomas *et al*, 2006; Barone *et al*, 2007; Lammers *et al*, 2008; Yan *et al*, 2011). Then, using *ex vivo* isolated duodenal epithelial cells to investigate how CRAMP may affect gliadin-triggered cellular signalling. We observed that CRAMP inhibited gliadin-triggered phosphorylation of EGFR at Tyr1068, Pi3K/Akt activation, MyD88/TNF receptor-associated factor 6 (TRAF6) expression and NF-κB activation in duodenal

epithelial cells *ex vivo*, while deficiency CRAMP intensified gliadin-triggered inflammation (Fig 4A). Similarly, an inhibitory effect of LL-37, the human ortholog of CRAMP, on pepsin/trypsin digested (PT)-gliadin-induced signalling was demonstrated on human epithelial cells (Fig 4B). To further determine the mechanism by which LL-37 inhibits gliadin-induced signalling, human epithelial cells were pre-treated with Ilomastat before stimulation with PT-gliadin. Ilomastat, an inhibitor of EGFR transactivation, was used to test whether LL-37 acts via a competitive binding (Tjabringa *et al*, 2003;

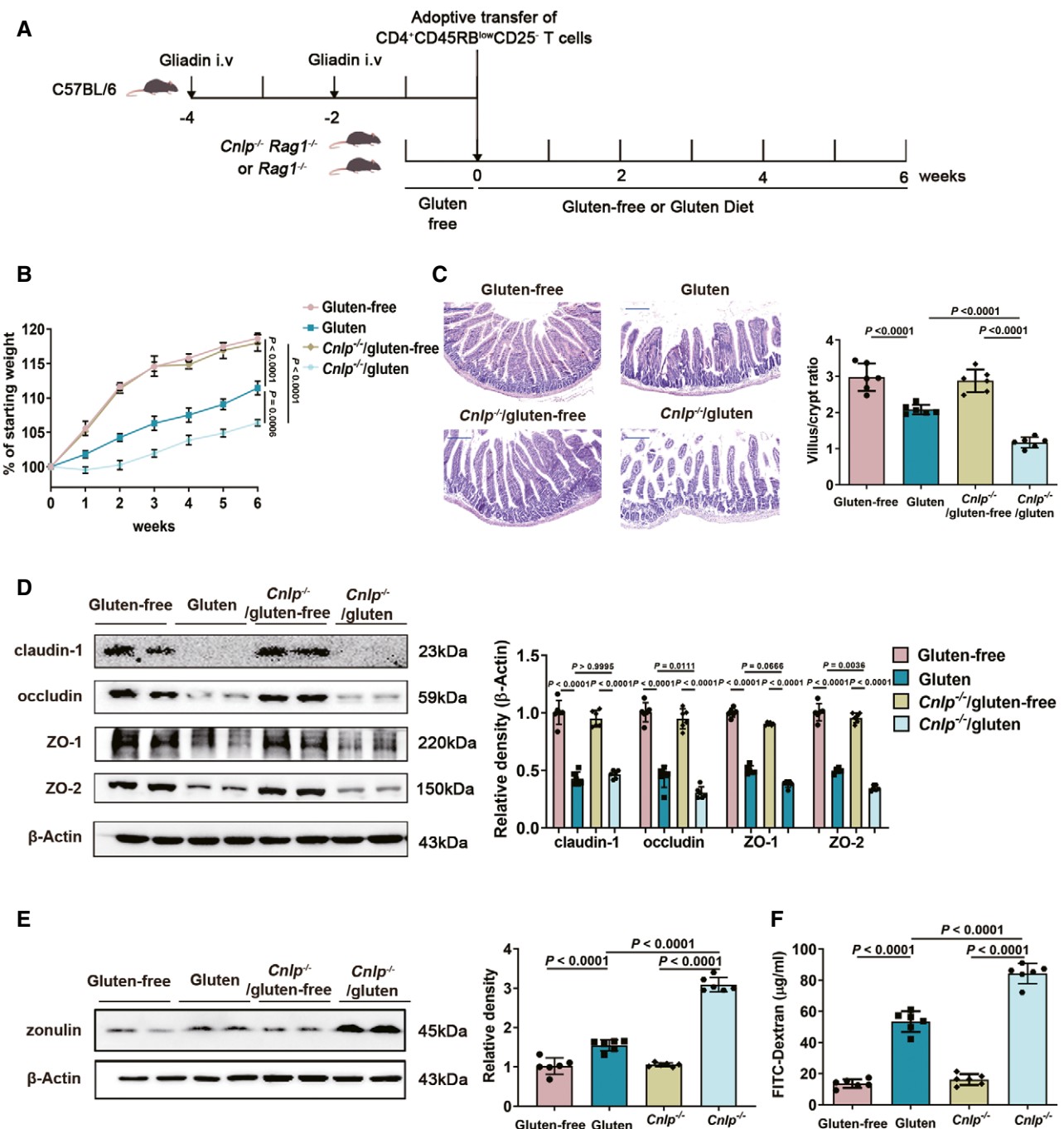

**Figure 2. CRAMP deficiency potentiates GIE.**

A   Animal protocol.

B   Changes in body weight relative to starting weight during 6 weeks (*n* = 6).

C   Left, representative images of duodenal damage by H&E staining. Scale bar: 200 µm. Right, graph depicted the ratio of the morphometric assessment of villus height to crypt depth (*n* = 6).

D   Western blot and densitometry analyses of duodenal tight junction proteins (claudin-1, occludin, ZO-1 and ZO-2; *n* = 6).

E   Western blot and densitometry analysis of duodenal zonulin (*n* = 6).

F   Intestinal permeability was assessed by measuring FITC-Dextran (*n* = 6).

Data information: Data (B–F) were representative and were the mean ± SD from three independent experiments. *P* values were calculated by one-way ANOVA followed by Tukey's *post hoc* test for multiple comparisons.

Source data are available online for this figure.

Sho *et al*, 2005). The results showed that the inhibitory effect of LL-37 on PT-gliadin-induced signalling was abolished by Ilomastat (Fig 4B), supporting that CRAMP counteracts gliadin-induced signalling by inducing a competitive binding to interrupt prolonged pathological activation of EGFR by gliadin.

### CRAMP modulates intestinal immune disorder in GIE

In addition to barrier dysfunction, gliadin also triggers epithelial cell to release interleukin-15 (IL-15) which importantly activates intraepithelial lymphocytes (IELs) (Villella *et al*, 2019). IL-15 activated IELs

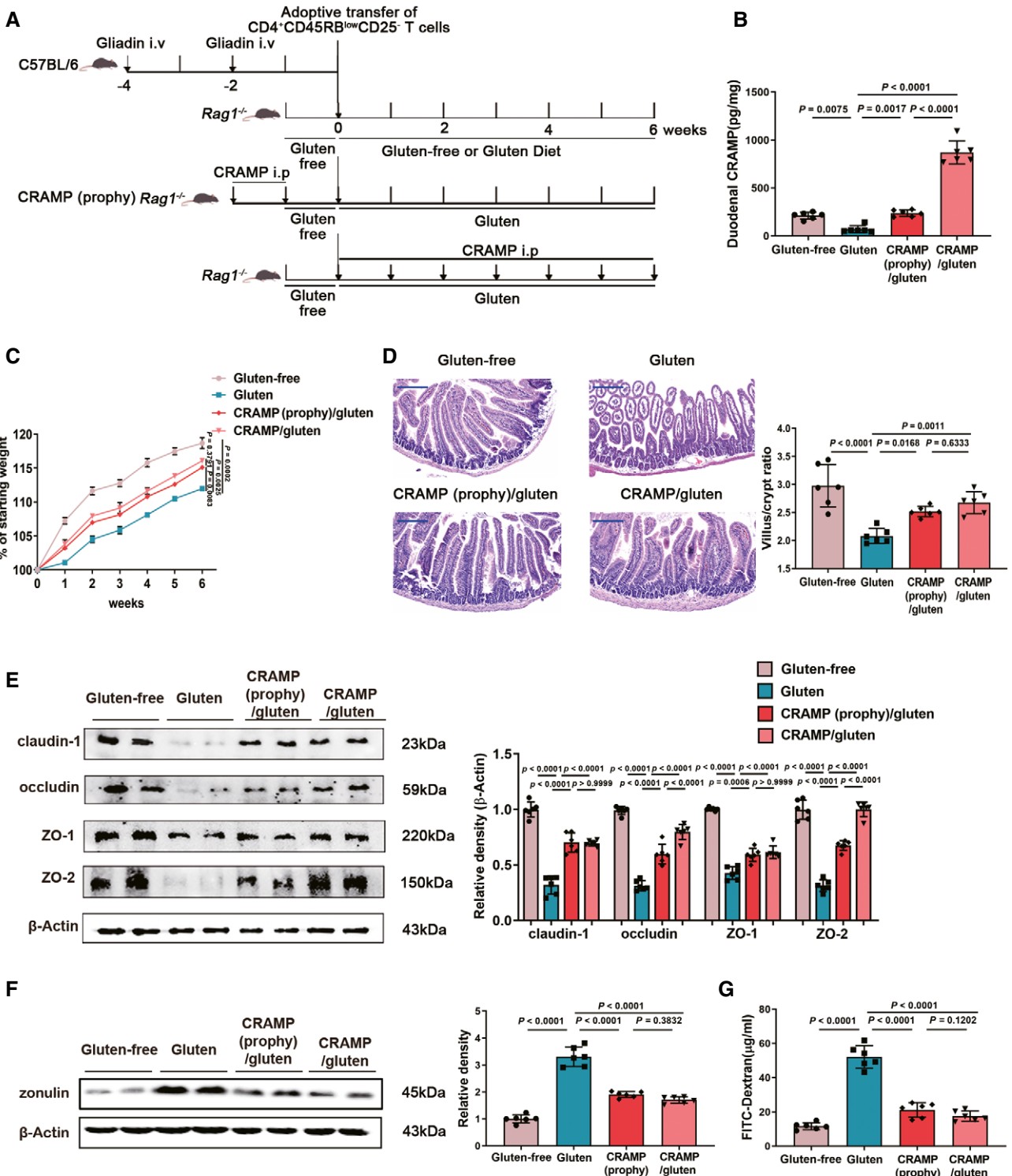

**Figure 3.**

**Figure 3.  Replenishing duodenal CRAMP is protective against GIE.**

A  Animal protocol.
B  Duodenal CRAMP determination by ELISA ($n = 6$).
C  Changes in body weight relative to starting weight during 6 weeks ($n = 6$).
D  Left, representative images of duodenal damage by H&E staining. Scale bar: 200 μm. Right, graph depicted the ratio of the morphometric assessment of villus height to crypt depth ($n = 6$).
E  Western blot and densitometry analyses of duodenal tight junction proteins (claudin-1, occludin, ZO-1 and ZO-2; $n = 6$).
F  Western blot and densitometry analysis of duodenal zonulin ($n = 6$).
G  Intestinal permeability was assessed by measuring FITC-Dextran ($n = 6$).

Data information: Data (B–G) were representative and were the mean ± SD from three independent experiments. *P* values were calculated by one-way ANOVA followed by Tukey's *post hoc* test for multiple comparisons.
Source data are available online for this figure.

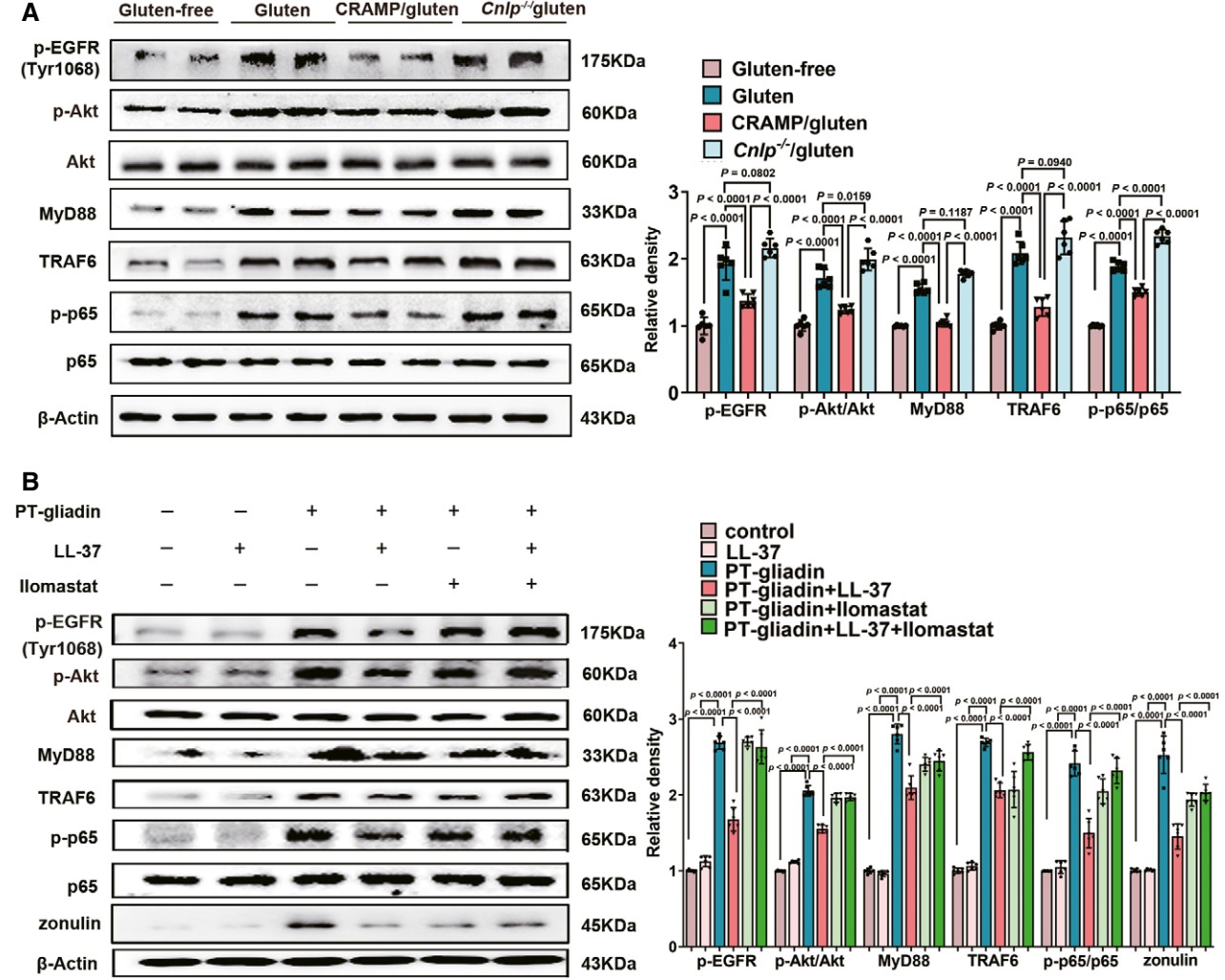

**Figure 4.  CRAMP counteracts gliadin-triggered inflammatory signalling to protect epithelial function.**

A  Western blot and densitometry analyses of gliadin-triggered inflammatory signalling [p-EGFR (Tyr1068), p-Akt, Akt, MyD88, TRAF6, p-p65 and p65] in epithelial cells *ex vivo* ($n = 6$).
B  Western blot and densitometry analyses of gliadin-triggered inflammatory signalling [p-EGFR (Tyr1068), p-Akt, Akt, MyD88, TRAF6, p-p65, p65 and zonulin] *in vitro* ($n = 6$).

Data information: Data (A and B) were representative and were the mean ± SD from three independent experiments. *P* values were calculated by one-way ANOVA followed by Tukey's *post hoc* test for multiple comparisons.
Source data are available online for this figure.

acquire cytotoxic properties by upregulating natural-killer group 2 member D (NKG2D) receptor, responsible for epithelial cell damage, leading to dysregulated immune responses at intestinal mucosa in celiac disease (Qingsheng *et al*, 2006; Valérie & Bana, 2014). We

examined whether CRAMP impacts on IL-15 production and mucosal immune imbalance during GIE. Therapeutic treatment of cathelicidin significantly reduced *Il15/IL15* expression in mouse duodenum and in human epithelial cells (Fig EV3A and B). *Ex vivo*

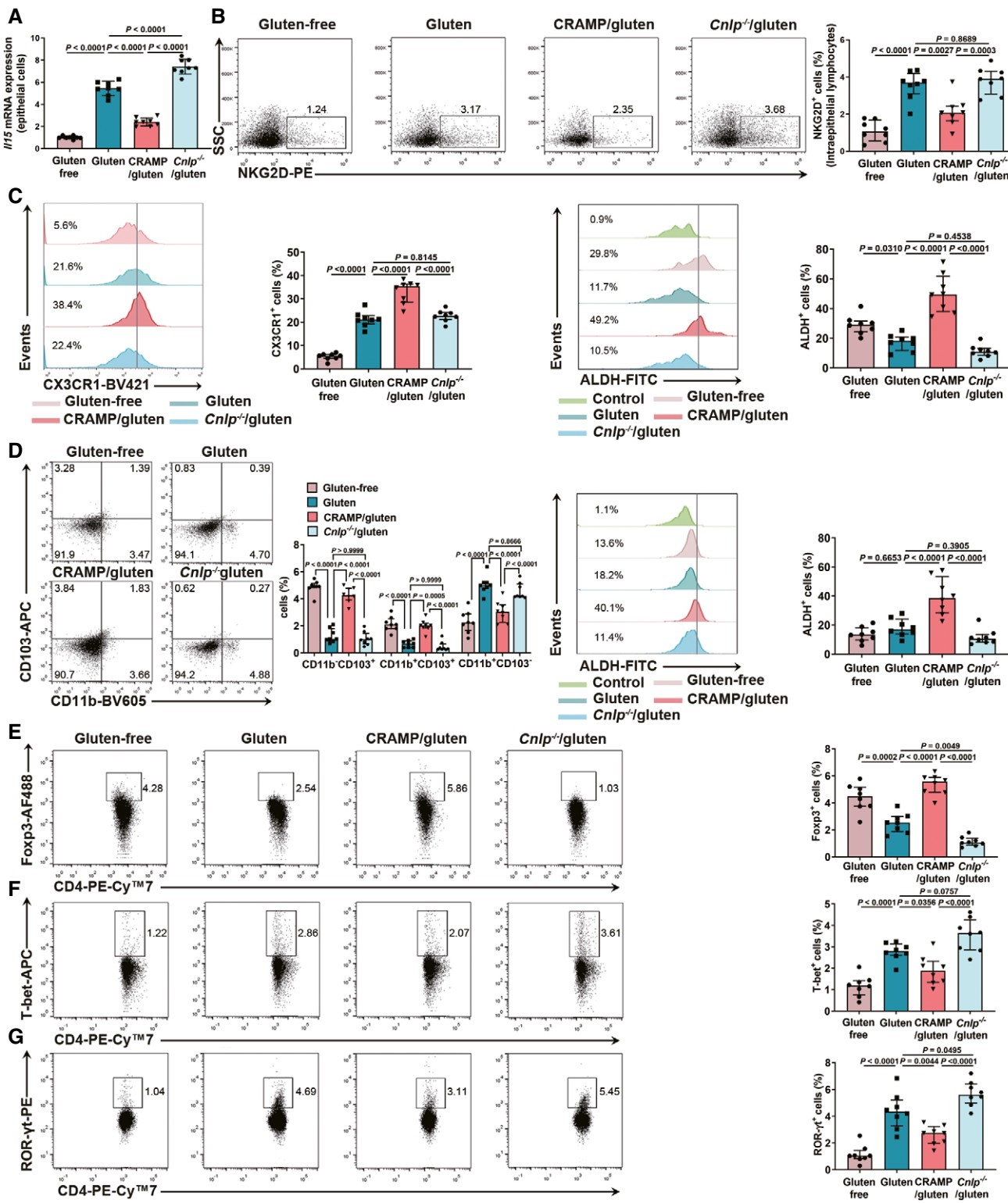

Figure 5.

**Figure 5. CRAMP modulates intestinal immune disorder in GIE.**

A   *Il15* expression in duodenal epithelial cells *ex vivo* (*n* = 8).

B   The frequency of NKG2D$^+$ IELs among CD3$^+$CD4$^+$ population (*n* = 8).

C   The frequency of CX3CR1$^+$ macrophages in CD45$^+$CD11b$^+$ population and ALDH activity in CD45$^+$CD11b$^+$CX3CR1$^+$ macrophages in lamina propria (*n* = 8).

D   The percentages of three subsets of CD45$^+$CD11c$^+$ DCs and ALDH activity in CD45$^+$CD11c$^+$ DCs in lamina propria (*n* = 8).

E–G   The percentages of (E) Foxp3$^+$ Treg, (F) T-bet$^+$ Th1 and (G) ROR$\gamma$t$^+$ Th17 among CD3$^+$CD4$^+$ population in lamina propria were shown by flow cytometry (*n* = 8).

Data information: Data in A were representative and were the mean ± SD from two independent experiments. Data (B–G) were representative and were the median ± interquartile range from three independent experiments. *P* values were calculated by one-way ANOVA followed by Tukey's *post hoc* test for multiple comparisons.

Source data are available online for this figure.

examination also confirmed that CRAMP downregulated IL-15 expression in duodenal epithelial cells (Fig 5A). Consequently, this led to attenuated percentage of NKG2D$^+$ IELs (Fig 5B). The regulatory effect of CRAMP on IL-15 production and cytotoxic IELs expansion was further confirmed by the observation that CRAMP-deficient mice with GIE (*Cnlp$^{-/-}$*/gluten group) exhibited a significantly increased percentage of NKG2D$^+$ IELs (Fig 5B).

Innate and adaptive immune cells in lamina propria (LP), including macrophages and dendritic cell (DCs) of different phenotypes as well as differentially polarized T cells, play important roles in development of celiac disease (Meresse *et al*, 2012). We observed that CRAMP induced CD45$^+$CD11b$^+$CX3CR1$^+$ macrophages and conventional CD45$^+$CD11c$^+$DCs (cDCs) exhibiting a high aldehyde dehydrogenase (ALDH) activity (Fig 5C and D). CRAMP treatment stimulated the numbers of regulatory CD11b$^-$CD103$^+$ and CD11b$^+$CD103$^+$DCs and decreased inflammatory CD11b$^+$CD103$^-$ DCs (Fig 5D). Accordingly, CRAMP treatment induced regulatory T cells (Treg) (Fig 5E) and reduced T helper 1 (Th1) (Fig 5F) and Th17 cells (Fig 5G). While restoring duodenal CRAMP harbours positive immunoregulatory effects, CRAMP deficiency further exacerbated gluten-induced immune dysregulation (Fig 5A–G). Collectively, CRAMP shapes intestinal mucosal immune environment by decreasing cytotoxic NKG2D$^+$ IELs and modulating DCs and macrophages phenotypes to induce regulatory T cells.

### Gut dysbiosis contributes to CRAMP degradation

Finally, we aimed to determine why CRAMP production in epithelium is defective in GIE. Our as well as other groups have demonstrated that CRAMP expression may be regulated by gut microbiota and metabolites such as short-chain fatty acids or proteases (Jan &

Pike, 2009; Sun *et al*, 2015). Patients with celiac disease have altered faecal and duodenal microbiota compositions compared to healthy individuals, which is partially restored after treatment with a gluten-free diet, suggesting the gut microbiota might play a vital role in celiac disease (Verdu *et al*, 2015). To test how GIE-associated dysbiosis contributes to defective CRAMP production, we analysed gliadin-induced perturbation of intestinal microbiota composition. Principal coordinates analysis (PCoA) showed that the gut microbiota communities in mice fed with gluten-containing diet were remarkably different from gluten-free diet group (Fig 6A). Analyses of microbiota composition at phylum level (Fig 6B), genus level (Fig 6C) and species level (Fig 6D) revealed that *P. aeruginosa*, an opportunistic pathogen unique to celiac disease patients and increased by gliadin (Caminero *et al*, 2019a), was increased, while *Akkermansia muciniphila* (*A. muciniphila*) was significantly decreased in gluten group. Butyrate-producing bacteria did not show any significant change during GIE (Fig 6C), neither did faecal levels of short-chain fatty acids (SCFAs) (Fig EV4). Pathogenic bacteria, including *P. aeruginosa*, are able to counteract host antimicrobial mechanism by producing proteases to cleave and inactivate cathelicidins (Jan & Pike, 2009; Cole & Nizet, 2016). Indeed, we found that the production of LasB, a protease of *P. aeruginosa,* was markedly increased in gluten group (Fig 6E). Moreover, at genus level, CRAMP was negatively associated with *P. aeruginosa* while positively correlated with *A. muciniphila* (Fig 6F). To confirm that *P. aeruginosa* could contribute to reduced CRAMP production, mice were orally supplemented with a highly virulent or a moderately virulent strain of *P. aeruginosa*, PA14 or PAO1 (Fig 6G and H). As shown in Fig 6I, reduced CRAMP production was observed in duodenum with colonization of the two *P. aeruginosa* strains, with a more pronounced effect observed with PA14. These data support

**Figure 6. Gut dysbiosis contributes to CRAMP degradation.**

A   The unweighted (left) and weighted (right) UniFrac distances based on OTU abundance (NC: gluten-free, M: gluten; *n* = 6).

B   The taxonomic composition distribution at phylum level (*n* = 6).

C   The taxonomic composition distribution at genus level (*n* = 6).

D   Duodenal colonizing *P. aeruginosa* and *A. muciniphila* as determined by RT–qPCR using bacterial-specific species gene primers (*n* = 6).

E   The production of LasB (a protease of *P. aeruginosa*) was examined in duodenum (*n* = 6).

F   Dot plot of Pearson correlation coefficients among CRAMP, *P. aeruginosa* and *A. muciniphila* at species level. The size of each point represented the correlation coefficient and the colour represented positive (red) or negative (purple) relationship (*n* = 6).

G   Animal protocol.

H   Duodenal colonizing PA14 and PAO1 as determined by RT–qPCR using *P. aeruginosa*-specific species gene primer (*n* = 9).

I   Western blot and densitometry analyses of pro-form CRAMP and mature peptide in duodenum (*n* = 8).

Data information: Data (D, E, H, I) were representative and were the mean ± SD from three independent experiments. *P* values were calculated by unpaired two-tailed *t*-test for comparison of two groups (B–E) or one-way ANOVA followed by Tukey's *post hoc* test for multiple comparisons (H, I).

Source data are available online for this figure.

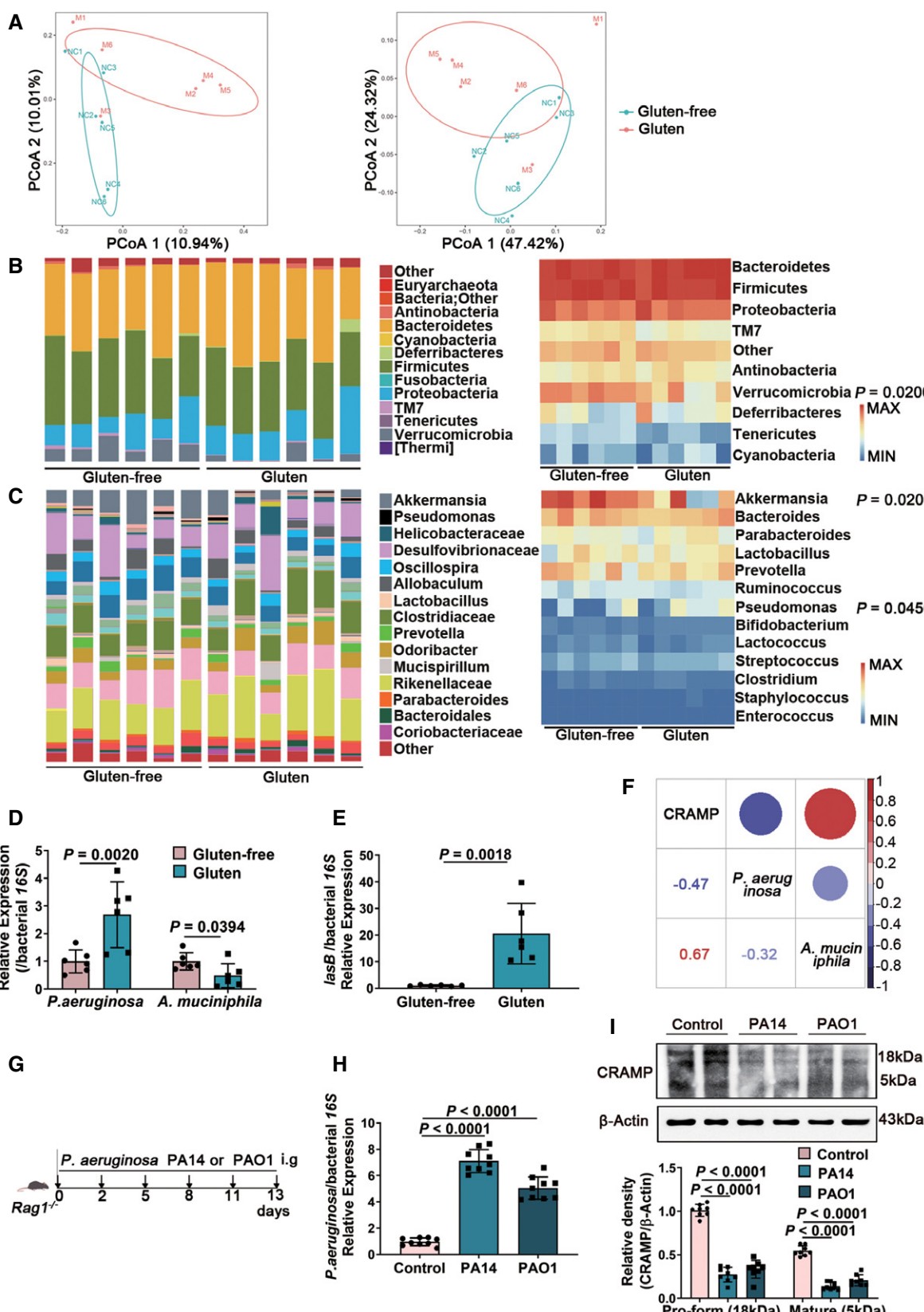

**Figure 6.**

that GIE-associated gut dysbiosis with increased *P. aeruginosa* contribute to defective CRAMP production.

### *Pseudomonas aeruginosa* depletion restores CRAMP production and dampens GIE

To finally confirm that gut dysbiosis favours defective CRAMP production and GIE development, we applied an antibiotic cocktail (ABX) to deplete mice of primary *P. aeruginosa* and increase *A. muciniphila* (Fig 7A) (Hill *et al*, 2010). We found that the ABX cocktail effectively reduced highly resistant pathogenic bacteria *P. aeruginosa* with an ability to cleave CRAMP, while keeping the beneficial bacteria, such as *A. muciniphila* intact (Fig EV5A–C), which consequently led to increased CRAMP production in serum (Fig 7B) and in duodenum (Fig 7C), supporting a direct effect of dysbiosis on duodenal CRAMP expression. Accordingly, ABX treatment alleviated the barrier damage (Fig 7D–F) and immune cell dysregulation in epithelium (Fig 7G) and lamina propria (Fig 7H–L) caused by gluten, reducing NKG2D$^+$ IELs and promoting modulatory phenotypes of macrophages, DCs and T cells. Together, our results support that gut microbiota via intestinal CRAMP production, impact mucosal barrier and immune environment and thus the development of GIE.

## Discussion

Our study supports that perturbation of gut microbiota composition during GIE contributes to defective production of epithelial CRAMP. CRAMP promotes epithelial barrier integrity and function by counteracting gliadin-triggered signalling. Additionally, CRAMP harbours positive immunoregulatory effects at intestinal mucosa by downregulating epithelial IL-15 and promoting modulatory phenotypes of macrophages and DCs to induce Treg differentiation. Thus, a dysregulated immune-gut microbiota axis triggered by gliadin results in low duodenal CRAMP levels, an unopposed mucosal barrier and immune dysregulation, and the development of GIE.

The mammalian gut epithelium produces a diverse collection of AMPs, to confront the complex microbial challenges. Although the small intestine is not considered as the primary site for cathelicidin production (Gallo & Hooper, 2012) compared to the colon, CRAMP is found to be expressed by small intestinal epithelial cells of adult mice (Gallo *et al*, 1997; Mukherjee & Hooper, 2015). Our data confirm its expression and secretion from epithelial cells in the duodenum of *Rag1*$^{-/-}$, BALB/c and C57BL/6 mice but not of CRAMP-deficient mice (Fig EV1A). In addition, host immune state has a profound impact on gut microbiota (Wu & Wu, 2012). Upon induction of GIE immunopathology, distorted gut microbiota composition was observed three weeks after adoptive transfer of gliadin-restricted T cells, accompanied by subsequently reduced CRAMP production, suggesting an immune-gut microbiota axis to regulate endogenous antimicrobial peptide.

Effects of gliadin on epithelial dysfunction during GIE may be mediated via prolonged EGFR activation or a MyD88-dependent mechanism to release zonulin (Thomas *et al*, 2006; Lammers *et al*, 2008). Gliadin triggered EGFR activation involves its phosphorylation at Tyr1068 (Barone *et al*, 2007), which leads to downstream Pi3K/Akt-NF-κB activation and inflammatory responses (Yan *et al*, 2011). Zonulin is the only protein known to date to reversibly regulate intestinal permeability by modulating intercellular TJPs (Amit *et al*, 2009). Gliadin-induced zonulin release is thought to be mediated via C-X-C motif chemokine receptor 3 (CXCR3) and MyD88 signalling (Lammers *et al*, 2008), which enhances intestinal permeability, a key pathologic event of GIE. Indeed, it has been suggested that gliadin acts by prolonging binding and activation of EGFR by its native ligand EGF and delays receptor inactivation by interfering its endocytic pathway (Barone *et al*, 2007). Interestingly, cathelicidins (CRAMP or LL-37) may also activate EGFR (Tjabringa *et al*, 2003; Sho *et al*, 2005; Niyonsaba *et al*, 2007; Sun *et al*, 2015) through a "trans-activation" mechanism involving EGFR phosphorylation at a different tyrosine site (Sho *et al*, 2005). EGFR phosphorylation at different loci is associated with different cellular responses (Avraham & Yarden, 2011). Gliadin-induced EGFR phosphorylation at Tyr1068 mediates intestinal dysfunction (Barone *et al*, 2007). In contrast, EGFR phosphorylation at Tyr100 induced by probiotic-derived protein protects the intestinal barrier (Shen *et al*, 2018). While the exact mechanism remains to be determined, cathelicidin-induced EGFR trans-activation has been shown to depend on metalloproteinase (MMP)-mediated HB-EGF shedding and binding to EGFR (Tjabringa *et al*, 2003), which is in agreement with our observation that the inhibitory effect of LL-37 on gliadin-induced EGFR phosphorylation (Tyr1068) was abolished by a MMP inhibitor Ilomastat. Thus, LL-37 counteracts gliadin-induced

---

**Figure 7. *Pseudomonas aeruginosa* depletion restores CRAMP production and dampens GIE.**

A    Animal protocol.
B    The levels of CRAMP in serum ($n = 8$).
C    The levels of CRAMP in duodenum ($n = 6$).
D    Representative images of intestinal damage by H&E staining. Scale bar: 200 μm. The graph depicted the ratio of the morphometric assessment of villus height to crypt depth ($n = 6$).
E    Western blot and densitometry analyses of tight junction proteins (claudin-1 and occludin) and zonulin in duodenum ($n = 6$).
F    Intestinal permeability was assessed by measuring FITC-Dextran ($n = 6$).
G    The frequency of NKG2D$^+$ IELs among CD3$^+$CD4$^+$ population ($n = 8$).
H    The frequency of CX3CR1$^+$ macrophages among CD45$^+$CD11b$^+$ population ($n = 8$).
I    The percentages of three subsets of CD45$^+$CD11c$^+$ DCs ($n = 8$).
J–L    The percentages of (J) Foxp3$^+$ Treg, (K) T-bet$^+$ Th1 and (L) RORγt$^+$ Th17 among CD3$^+$CD4$^+$ population in lamina propria were shown by flow cytometry ($n = 8$).

Data information: Data (B–F) were representative and were the mean ± SD from three independent experiments. Data (G–L) were representative and were the median ± interquartile range from three independent experiments. *P* values were calculated by one-way ANOVA followed by Tukey's *post hoc* test for multiple comparisons. Source data are available online for this figure.

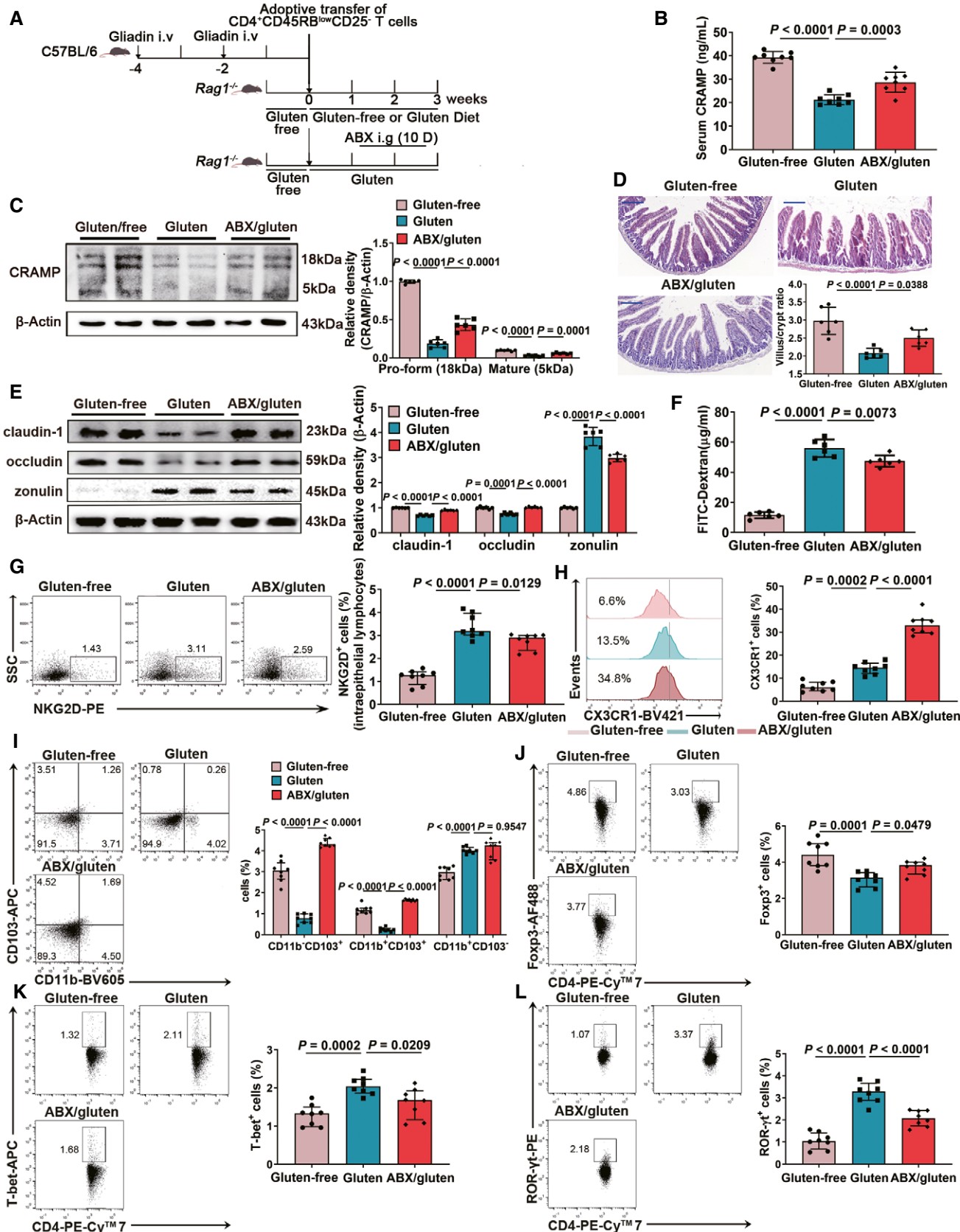

**Figure 7.**

signalling by inducing a competitive binding to interrupt prolonged pathological activation of EGFR by gliadin. In addition, earlier evidence has shown that LL-37 inhibits MyD88 expression (Pinheiro *et al*, 2009). This agrees with our data that LL-37 inhibits gliadin-induced MyD88 expression, thus contributing to reduced zonulin production. Together, it may be postulated that LL-37, by preventing gliadin-induced EGFR Tyr1068-phosphorylation associated with pathologic consequences and by inhibiting MyD88-dependent zonulin, protects against gliadin-induced epithelial dysfunction in GIE.

Another hallmark of celiac disease is NF-κB-regulated IL-15 expression. IL-15 may be released from epithelium upon its stimulation by gliadin (Valérie & Bana, 2014) and plays a critical role in IEL-mediated epithelial cell damage and in dysregulated immune responses during celiac disease. IL-15 is transcriptionally regulated by NF-κB. The p65 subunit of NF-κB binds to the IL-15 NF-κB motif (Villella *et al*, 2019). Additional, MyD88-dependent signalling is important for regulation of IL-15 production and in maintenance of IELs (Qingsheng *et al*, 2006). Cathelicidins have been shown to suppress the translocation of NF-κB subunits (Mookherjee *et al*, 2006) as well as MyD88 expression. Consistently, we observed that cathelicidin antagonized gliadin-induced NF-κB activation and MyD88 signalling and suppressed epithelial IL-15 production. Following its release from epithelial cells, IL-15 activates intraepithelial lymphocytes responsible for epithelial cell damage (Valérie & Bana, 2014). Furthermore, IL-15 treatment reversed the effect of retinoic acid (RA) by altering the tolerogenic phenotype of intestinal DCs, hence preventing the generation of inducible Treg cells to dietary gluten and promoting the development of Th1 and Th17 inflammatory immune responses (Valérie & Bana, 2014). Here, our data support that CRAMP reduces epithelial toxic NKG2D$^+$ IELs by IL-15 inhibition. Additionally, our data demonstrate that CRAMP induces ALDH$^+$ macrophages and ALDH$^+$ cDCs. ALDH is an important enzyme present in innate immune cells important for the production of RA, which is a key mechanism of these cells to induce Treg cells (Guilliams *et al*, 2010) and to inhibit the development of Th17 (Daniel *et al*, 2007). In addition, at intestinal mucosa, cDCs may be subdivided into three subsets based on the expression of CD11b and CD103: CD11b$^-$CD103$^+$, CD11b$^+$CD103$^-$ and CD11b$^+$CD103$^+$ DCs, exhibiting inflammatory or modulatory properties. The CD11b$^+$ CD103$^+$DCs are so far found to be unique to the intestines and to preferentially drive the development of Foxp3$^+$ Treg, which is dependent on RA (Denning *et al*, 2007; Sun *et al*, 2007). We observe that CRAMP induces the regulatory subsets of cDCs (CD11b$^-$CD103$^+$ and CD11b$^+$CD103$^+$) and reduces the inflammatory subsets (CD11b$^+$CD103$^-$), suggesting its additional immune modulatory mechanism. All these mechanisms contribute to regulatory T cells polarization induced by CRAMP.

Here, we demonstrate that a dysregulated immune-gut microbiota axis contributes to epithelial CRAMP loss. To be noted, cathelicidin antimicrobial peptide is part of host defence mechanisms to exhibit broad-spectrum activities against microbial invasions. Nevertheless, during co-evolution with the host, opportunistic pathogens have also developed a variety of means to resist AMPs (Nizet, 2006). For examples, proteases of *P. aeruginosa*, *Staphylococcus aureus*, *Streptococcus pyogenes*, *Porphyromonas gingivalis* and *Enterococcus faecalis* lead to degradation

of LL-37 and consequently abolish antimicrobial activity (Jan & Pike, 2009). Celiac disease is associated with typical gut dysbiosis (Sanchez *et al*, 2013; Verdu *et al*, 2015), and the role of gut microbiota appears to be complex. Mice colonized with a microbiota devoid of opportunistic pathogens develop attenuated dietary gluten-induced immunopathology compared with germ-free mice and mice that harbour a complex microbiota that includes opportunistic pathogens (Galipeau *et al*, 2015). Specific gut bacteria may promote or protect celiac disease-associated immunopathology (Caminero *et al*, 2019a; Caminero *et al*, 2019b). We observed that gut dysbiosis developed with induction of GIE, with increased *P. aeruginosa* and reduced *A. muciniphila*, which were similarly observed in clinical celiac disease (Bodkhe *et al*, 2019; Caminero *et al*, 2019a). Duodenal tissue was detected with increased abundance of *P. aeruginosa*, confirming its ability of translocating from colon to duodenum. Recently, the pathogenic mechanism of *P. aeruginosa* in celiac disease has been revealed (Caminero *et al*, 2016; Caminero *et al*, 2019a). In mice expressing celiac disease risk genes, the metabolite elastase, also called LasB of *P. aeruginosa* synergizes with gluten to induce severe inflammation (Caminero *et al*, 2019a). It is suggested that *P. aeruginosa*-derived LasB promotes the propagation of celiac disease mediated by the protease-activated receptor 2 (PAR2) pathway, which interacts with zonulin (Amit *et al*, 2009; Caminero *et al*, 2019a). Also, *P. aeruginosa* produces LasB with demonstrated ability to degrade CRAMP (Jan & Pike, 2009). Indeed, we observed increased LasB in duodenal tissue, corresponding to reduced CRAMP. We further confirmed this by applying antibiotic treatment able to deplete *P. aeruginosa*. Such treatment dampened the development of GIE, by preventing *P. aeruginosa*-mediated degradation of CRAMP, confirming the role of gut dysbiosis in defective CRAMP production. The effect of antibiotic therapy in the development of GIE is complex, dependent on its targeted gut bacteria. Accordingly, selective antibiotic treatment targeting different bacterial subgroups may yield varying outcome on celiac disease (Hansen *et al*, 2013; Dydensborg Sander *et al*, 2019). Perinatal antibiotic treatment, which led to increased numbers of *Proteobacteria*, enhanced the severity of GIE. Vancomycin treatment that propagates *A. muciniphila*, reduces interferon-gamma (IFN-γ) and IL-15 levels in the intestine and NKG2D ligand expression on intestinal epithelial cells (IECs), thus protecting GIE (Hansen *et al*, 2013). While *P. aeruginosa* is resistant to most antibiotics (Breidenstein *et al*, 2011), our combinatory antibiotics effectively reduce *P. aeruginosa*, which is attributable for its protective effect on GIE.

Altogether, in current GIE model we demonstrate that the undigested immunogenic gliadin from gluten triggers a dysregulated immune-gut microbiota axis that negatively impacts intestinal CRAMP production, consequently inducing pathological dysregulation of barrier function and immune environment at intestinal mucosa and unopposed development of GIE. It should be interesting to determine in other experimental models of celiac disease and in susceptible individuals whether similar effect and mechanism of CRAMP exist. Finally, our data support that manipulation of gut microbiota and the production of intestinal AMPs may represent an attractive therapeutic strategy to maintain intestinal homeostasis and prevent the development of celiac disease.

# Materials and Methods

## Reagents and Tools table

| Reagent/Resource | Reference or Source | Identifier or Catalog Number |
|---|---|---|
| **Experimental Models** | | |
| Mouse: C57BL/6J: | GemPharmatech | C57BL/6JGpt; Cat# N000295; RRID: SCR_017239 |
| Mouse: *Cnlp*$^{-/-}$: | Jackson Laboratory | B6.129X1-Camp$^{tm1Rlg}$/J Mus musculus; Cat# JAX:017799; RRID: IMSR_JAX:017799 |
| Mouse: *Rag1*$^{-/-}$: | GemPharmatech | B6/JGpt-*Rag1*$^{em1Cd}$/Gpt; Cat# T004753; RRID: SCR_017239 |
| Mouse: *Cnlp*$^{-/-}$*Rag1*$^{-/-}$ | This study | N/A |
| Cell lines: Caco-2 | ATCC | Cat# HTB-37; RRID: CVCL_0025 |
| Patient samples: Human serum | Second Affiliated Hospital of Nanchang University | Ethics number: no. [2010] 041 |
| Bacterial strains: *P. aeruginosa* PAO1 | ATCC | Cat# 47085 |
| Bacterial strains: *P. aeruginosa* PA14 | Isolated from patient | N/A |
| **Antibodies** | | |
| Anti-CRAMP (1-39) antibody | Innovagen | Cat# PA-CRPL-100 |
| Anti-claudin-1 antibody | Abcam | Cat# ab180158 |
| Anti-occludin antibody | Abcam | Cat# ab216327; RRID: AB_2737295 |
| Anti-zonulin antibody | Abcam | Cat# ab131236; RRID: AB_11157376 |
| Anti-TRAF6 antibody | Abcam | Cat# ab40675; RRID: AB_778573 |
| Anti-p65 antibody | Abcam | Cat# ab32536; RRID: AB_776751 |
| Anti-p-AKT antibody | Cell Signaling Technology | Cat# 4060; RRID: AB_2315049 |
| Anti-AKT antibody | Cell Signaling Technology | Cat# 4685; RRID: AB_2225340 |
| Anti-MyD88 antibody | Cell Signaling Technology | Cat# 4283; RRID: AB_10547882 |
| Anti-p-p65 antibody | Cell Signaling Technology | Cat# 3033; RRID: AB_331284 |
| Anti-ZO-1 antibody | Thermo Fisher Scientific | Cat# 40-2200; RRID: AB_2533456 |
| Anti-ZO-2 antibody | Thermo Fisher Scientific | Cat# 71-1400; RRID: AB_88012 |
| Anti-ACTB antibody | ABclonal | Cat# AC026; RRID: AB_2768234 |
| Alexa Fluor® 700 anti-mouse CD45 | Biolegend | Cat# 103128; RRID: AB_493715 |
| Brilliant Violet® 711 anti-mouse CD11c | Biolegend | Cat# 117349; RRID: AB_2563905 |
| Brilliant Violet® 605 anti-mouse CD11b | Biolegend | Cat# 117349; RRID: AB_2563905 |
| APC anti-mouse CD103 | Biolegend | Cat# 12141; RRID: AB_1227503 |
| Brilliant Violet® 421 anti-mouse CX3CR1 | Biolegend | Cat# 149023; RRID: AB_2565706 |
| Brilliant Violet® 421 anti-mouse CD3e | Biolegend | Cat# 100341; RRID: AB_2562556 |
| PE anti-mouse CD314 (NKG2D) | Biolegend | Cat# 130207; RRID: AB_1227713 |
| APC anti-T-bet | Biolegend | Cat# 644813; RRID: AB_10896913 |
| Alexa Fluor® 488 anti-mouse FOXP3 | Biolegend | Cat# 126405; RRID: AB_1089114 |
| PE anti-mouse ROR GAMMA (T) (B2D) | Thermo Fisher Scientific | Cat# 12-6981-80; RRID: AB_10805392 |
| PE-Cy™7 anti-mouse CD4 | BD biosciences | Cat# 552775; RRID: AB_394461 |
| anti-CRAMP pAb | Proteintech | Cat# 12009-1-AP; RRID: AB_908736 |

**Reagents and Tools table** (continued)

| Reagent/Resource | Reference or Source | Identifier or Catalog Number |
|---|---|---|
| anti-E-cadherin pAb | BD biosciences | Cat# 610181; RRID: AB_397580 |
| anti-F4/80 mAb | Abcam | Cat# ab6640; RRID: AB_1140040 |
| anti-Ly6G mAb | Abcam | Cat# ab25377; RRID: AB_470492 |
| CD8a Antibody, Biotin | Miltenyi Biotec | Cat# 130-118-147; RRID: AB_2733536 |
| CD11b Antibody, Biotin | Miltenyi Biotec | Cat# 130-113-233; RRID: AB_2726044 |
| CD11c Antibody, Biotin | Miltenyi Biotec | Cat# 130-113-578; RRID: AB_2726178 |
| CD19 Antibody, Biotin | Miltenyi Biotec | Cat# 130-113-729; RRID: AB_2726270 |
| CD45R(B220) Antibody, Biotin | Miltenyi Biotec | Cat# 130-123-853; RRID: AB_2819526 |
| CD49b(DX5) Antibody, Biotin | Miltenyi Biotec | Cat# 130-101-934; RRID: AB_2660466 |
| CD105 Antibody, Biotin | Miltenyi Biotec | Cat# 130-101-992; RRID: AB_2660100 |
| MHC Class II Antibody, Biotin | Miltenyi Biotec | Cat# 130-101-849; RRID: AB_2660066 |
| Ter-119 Antibody, Biotin | Miltenyi Biotec | Cat# 130-120-828; RRID: AB_2784480 |
| TCR$\gamma/\delta$ Antibody, Biotin | Miltenyi Biotec | Cat# 130-114-028; RRID: AB_2733574 |
| CD25 Antibody, Biotin | Miltenyi Biotec | Cat# 130-092-569; RRID: AB_871645 |
| Streptavidin MicroBeads | Miltenyi Biotec | Cat# 130-048-101 |
| CD45RB Antibody, FITC | Miltenyi Biotec | Cat# 130-102-461; RRID: AB_2658357 |
| Anti-FITC MicroBeads antibody | Miltenyi Biotec | Cat# 130-048-701; RRID: AB_244371 |
| **Oligonucleotides and other sequence-based reagents** | | |
| PCR primers | This study | Table EV2 |
| **Chemicals, Enzymes and other reagents** | | |
| CRAMP | GL biochem | Cat#088328 |
| Gliadin from wheat | Sigma | Cat#G3375-25G |
| Gluten from wheat | Sigma | Cat#G5004-500G |
| **Software** | | |
| FlowJo | BD Bioscience | RRID: SCR_008520; https://www.flowjo.com/solutions/flowjo/downloads |
| GraphPad Prism | GraphPad | RRID: SCR_002798; https://www.graphpad.com/ |
| Image J | Image J | RRID:SCR_003070; https://imagej.en.softonic.com/ |
| R project | R Project | RRID:SCR_001905; https://www.r-project.org/ |
| FluorChem FC3 | ProteinSimple | RRID:SCR_013724; http://www.proteinsimple.com/fluorchem_e.html |
| ZEN | Zeiss | RRID:SCR_018163; http://stmichaelshospitalresearch.ca/wp-content/uploads/2015/09/ZEN-Black-Quick-Guide.pdf |
| **Other** | | |
| Illumina MiSeq | Illumina | RRID:SCR_010233 http://www.illumina.com |
| CRAMP ELISA kit | My biosource | Cat# MBS7700441 |
| LL-37 ELISA kit | My biosource | Cat# MBS3800861 |

## Methods and Protocols

### Animals and treatments

All animal experimental protocols were approved by the Animal Ethics Committee of Jiangnan University [JN. No20181230c0500815 (303) and JN. No20170614-20190225 (77)] and were performed in accordance with the guidelines. Animal studies are reported in compliance with the ARRIVE guidelines (Kilkenny, 2010). Male C57BL/6 mice, *Rag1*$^{-/-}$ mice (N000295; T004753; GemPharmatech Co., Ltd, Jiangsu, China), CRAMP-deficient *Cnlp*$^{-/-}$ mice (C57BL/6 background; JAX:017799; The Jackson Laboratory, CA, USA) and *Cnlp*$^{-/-}$*Rag1*$^{-/-}$ mice were maintained at the Animal Housing Unit of Jiangnan University (Jiangsu, China) under a controlled temperature (23−25°C) and a 12 h light–12 h dark cycle. The *Cnlp*$^{-/-}$*Rag1*$^{-/-}$ mice were generated by backcrossing of *Cnlp*$^{-/-}$ mice with *Rag1*$^{-/-}$ mice for more than 10 generations. All experiments were performed with mice at age 6–8 weeks.

To recapitulate gliadin-induced small intestinal enteropathy and the gliadin-specific T-cell responses in active celiac disease patients, we chose an experimental model of dietary gluten-induced

enteropathy by adoptive gliadin-specific T cells transfer as previously described (Freitag *et al*, 2009). Briefly, C57BL/6 donor mice were maintained on a gluten-free standardized diet (AIN-76A; Research Diets, NJ, USA) from birth. Mice were given 100 μg gliadin (G3375-25G) or ovalbumin (negative control antigen) in Complete Freund's Adjuvant at tail base on day 0 and 50 μg gliadin or ovalbumin in Incomplete Freund's Adjuvant on day 14 (all Sigma, Shanghai, China). $Rag1^{-/-}$ and $Cnlp^{-/-}Rag1^{-/-}$ recipient mice were changed to AIN-76A on day 21. For sensitization, on day 28, $Rag1^{-/-}$ and $Cnlp^{-/-}Rag1^{-/-}$ recipient mice were injected intraperitoneally with $4.5 \times 10^5$ splenic $CD4^+CD45RB^{low}CD25^-$ T cells from C57BL/6 donor mice. Mice were divided randomly into experimental groups by different diet: gluten-free diet (AIN-76A) and gluten-containing diet [based on AIN-76A, containing 2.5 g wheat gluten (G5004-500G; Sigma, Shanghai, China) per kg] ($n$ = 6-8). Unsensitized mice did not receive splenic $CD4^+CD45RB^{low}CD25^-$ T cells and were fed with either gluten-free diet or gluten-containing diet. CRAMP (the mature form; 088328; GL biochem, Shanghai, China) was intraperitoneal administered (100 μg mouse$^{-1}$ week$^{-1}$) either two weeks before (prophylactic treatment) or 6 weeks after (therapeutic treatment) received adoptive T cells transfer. For antibiotic treatment (Hill *et al*, 2010), three weeks after adaptive transfer, mice were given 1 g l$^{-1}$ metronidazole, 1 g l$^{-1}$ gentamicin, 0.5 g l$^{-1}$ vancomycin, 1 g l$^{-1}$ ampicillin and 1 g l$^{-1}$ neomycin (all in Sigma, Shanghai, China) by daily oral gavage of 200 μl in antibiotic solution (100% H$_2$O) for 10 days (Hill *et al*, 2010). Mice were sacrificed by a lethal dose of pentobarbital sodium (90 mg kg$^{-1}$; Sigma, Shanghai, China).

### Human samples

This study was approved by the ethics committee of the Second Affiliated Hospital of Nanchang University [no. (2010) 041]. All subjects provided informed consent before participation and the experiments conformed to the principles set out in the WMA Declaration of Helsinki and the Department of Health and Human Services Belmont Report. The characteristics of participants were given in Appendix Table S1.

### Cell culture and treatment

Human intestinal epithelial cells Caco-2 (HTB-37) were obtained from the American Type Culture Collection (ATCC, VA, USA) and maintained in DMEM containing 1,800 mg l$^{-1}$ NaHCO$_3$, supplemented with 10% FBS, 100 U ml$^{-1}$ penicillin and 100 μg ml$^{-1}$ streptomycin at 37°C in a humidified atmosphere with 5% CO$_2$. LL-37 (GL biochem, Shanghai, China), a broad-spectrum matrix metalloproteinase inhibitor Ilomastat (Selleck, Shanghai, China) and PT-gliadin were dissolved in distilled water. For treatment, cells were pre-incubated with Ilomastat (5 μM) for 1 h, subsequently stimulated with PT-gliadin (1 mg ml$^{-1}$) and LL-37 (10 μg ml$^{-1}$) for 3 h. Cells were then collected for measurements. All cell lines were regularly tested negative for mycoplasma contamination.

### Bacterial supplementations

*Pseudomonas aeruginosa* PA14 (patient isolate) and PAO1 (47085; ATCC, WA, USA) were synthesized and kindly provided by Dr Lvyan Ma's laboratory in Chinese Academy of Sciences (Beijing, China). $Rag1^{-/-}$ mice were supplemented by oral gavage with *P. aeruginosa* PA14 or PAO1 three times a week for 2 weeks ($10^{10}$ cfu mouse$^{-1}$).

### The paper explained

#### Problem

Celiac disease is a complex immune-mediated small intestinal enteropathy, for which curative therapeutics are not available. Cathelicidin-related antimicrobial peptide (CRAMP) plays an indispensable role in intestinal homeostasis. However, the effect of CRAMP in dietary gluten-induced small intestinal enteropathy (GIE) such as celiac disease remains to be explored.

#### Results

Defective CRAMP production in duodenal epithelium was correlated with GIE development. CRAMP deficiency aggravated GIE, while exogenous CRAMP treatment protected against it. Furthermore, CRAMP harboured positive modulatory effect on intestinal barrier integrity by counteracting gliadin-triggered epithelial dysfunction and promoted modulatory immune cell responses at the intestinal mucosa. Lastly, GIE-associated enrichment of *Pseudomonas aeruginosa* and increased production of protease LasB contributed to defective intestinal CRAMP production.

#### Impact

This study provides novel evidence that CRAMP is a key regulatory mediator for GIE. Understanding the impact of gut microbiota-CRAMP axis on intestinal barrier integrity and immune responses during GIE may shed light on novel therapeutic strategies for clinical celiac disease.

### RNA isolation and reverse transcription real-time quantitative polymerase chain reaction (RT–qPCR)

Total RNA was extracted from tissues and cells using TRIzol following the manufacturer's protocol, and cDNAs were synthesized by a reverse transcription reagent kit (RR036A; TaKaRa, Kyoto, Japan). Gene expression levels were analysed by RT–qPCR using the Bio-Rad CFX Connect Real-Time System (CA, USA). After the preparation of the duodenal content (bacterial DNA extraction), the quantity of total bacteria, *P. aeruginosa* and *A. muciniphila* at species level were determined using bacterial-specific species gene primers and bacterial 16S gene sequence universal primer. Primer sequences were given in Appendix Table S2.

### Preparation of T cells fractions for adoptive transfer experiments

$CD4^+CD45RB^{low}CD25^-$ subset T cells were isolated from spleens as described (Mottet *et al*, 2003). In brief, single cell suspensions were stained with biotinylated anti-CD8a (130-118-147), CD11b (130-113-233), CD11c (130-113-578), CD19 (130-113-729), CD45R(B220) (130-123-853), CD49b(DX5) (130-101-934), CD105 (130-101-992), MHC Class II (130-101-849), Ter-119 (130-120-828), TCRγ/δ (130-114-028) and CD25 (130-092-569) followed by streptavidin MACS beads (130-048-101) and sorted on an AutoMACS (all Miltenyi Biotec, Bergisch Gladbach, Germany). The $CD4^+CD25^-$ fraction was then stained with anti-CD45RB-FITC (130-102-461), followed by incubation with anti-FITC MACS beads (130-048-701) and sorted on an AutoMACS.

### Statistical analyses

Data were expressed as mean $\pm$ SD. $P < 0.05$ was considered statistically significant. Difference between two groups was determined

using unpaired two-tailed *t*-test. Differences among three or more groups were determined analysis of variance (ANOVA) followed by Tukey's *post hoc* test. For flow cytometry statistical analyses, data were expressed as median ± interquartile range. The Pearson correlation coefficients between CRAMP and microbiota were analysed by the R Project (NJ, USA). All data were analysed using GraphPad Prism 8 software (CA, USA).

## Data availability

Mouse gut microbiota sequencing and assembly: Sequence Read Archive PRJNA686187 (https://www.ncbi.nlm.nih.gov/bioproject/?term=prjna686187).

The main data supporting the findings of this study are available within the article and its Expanded View Figures. Extra data are available from the corresponding author upon request.

**Expanded View** for this article is available online.

## Acknowledgements

We would like to thank Prof Lvyan Ma from Chinese Academy of Sciences (Beijing, China) for kindly providing two strains of *P. aeruginosa* PA14 and PAO1. The work was supported by funds from the National Natural Science Foundation of China (Grant Nos: 80270666, 81870439, 81973322, 91642114, 31570915, and National Youth 1000 Talents Plan), the Natural Science Foundation for Distinguished Young Scholars of Jiangsu Province (Grants No.: BK20200026), Jiangsu Province Recruitment Plan for High-level, Innovative and Entrepreneurial Talents (Innovative Research Team), Wuxi Social Development Funds for International Science & Technology Cooperation (Grant No: WX0303B010518180007PB), Jiangsu Province "Six Summit Talents" programme (YY-038), Jiangsu Province Qing Lan Project, National First-class Discipline Program of Food Science and Technology (Grant No: JUFSTR20180103), the Fundamental Research Funds for the Central Universities (Grant Nos.: JUSRP221037, JUSRP22007), Postgraduate Research & Practice Innovation Program of Jiangsu Province (Grant No: KYCX20_1876), Collaborative Innovation Center of Food Safety and Quality Control in Jiangsu Province and Wuxi Taihu Talent Project.

## Author contributions

ZR, L-LP and YH performed experiments and analysed data. HC and YL helped with obtaining the human samples. HL, XT, YL, BL, XD and XP assisted the experiments. YF and HFL helped with bacteria culture. BA and JD contributed to the data acquisition and critically reviewed the manuscript. ZR, JS and JD designed and interpreted experiments. JS, ZR and L-LP wrote the paper.

## Conflict of interest

The authors declare that they have no conflict of interest.

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
