## [Review Process File · EMBO Molecular Medicine]

Gut microbiota-CRAMP axis shapes intestinal barrier function and immune responses in dietary gluten-induced enteropathy

Zhengnan Ren, Li-Long Pan, Yiwen Huang, Yu Liu, Hongbing Chen, He Liu, Xing Tu, Yanyan Liu, Binbin Li, Xiaoliang Dong, Xiaohua Pan, Hanfei Li, Yu Fu, Birgitta Agerberth, Julien DIANA, and Jia Sun

DOI: [10.15252/emmm.202114059](https://doi.org/10.15252/emmm.202114059)

Corresponding authors: Jia Sun (jjasun@jiangnan.edu.cn) , Julien DIANA (julien.diana@inserm.fr)

Review Timeline:

Submission Date:	1st Feb 21
Editorial Decision:	16th Mar 21
Revision Received:	26th Apr 21
Editorial Decision:	12th May 21
Revision Received:	19th May 21
Accepted:	20th May 21

Editor: Zeljko Durdevic

Transaction Report:

16th Mar 2021

Dear Prof. Sun,

Thank you for the submission of your manuscript to EMBO Molecular Medicine, and please accept my apologies for the delay in getting back to you. We have received feedback from two of the three reviewers who agreed to evaluate your manuscript. Should referee #3 provide a report, we will send it to you, with the understanding that we will not ask for an additional revision. As you will see from the reports below, both referees find the study interesting and important. However, they also raise important criticism that I would like you to address in a major revision of the current manuscript.

Addressing the reviewers' concerns in full will be necessary for further considering the manuscript in our journal. Please note that EMBO Molecular Medicine encourages a single round of revision only and therefore, acceptance or rejection of the manuscript will depend on the completeness of your responses included in the next, final version of the manuscript. For this reason, and to save you from any frustrations in the end, I would strongly advise against returning an incomplete revision.

We would welcome the submission of a revised version within three months for further consideration. However, we realize that the current situation is exceptional on the account of the COVID-19/SARS-CoV-2 pandemic. Please let us know if you require longer to complete the revision.

I look forward to receiving your revised manuscript.

Yours sincerely,

Zeljko Durdevic

***** Reviewer's comments *****

Referee #1 (Comments on Novelty/Model System for Author):

This manuscript by Ren and colleagues investigates the function of gut microbiota-CRAMP-mediated modulation of intestinal barrier function and immune responses during gluten-induced enteropathy. The authors find that mice with GIE have altered microbiota compositions, which contributes to the CRAMP degradation in intestinal epithelium. Most intriguingly, they report that exogenous CRAMP treatment markedly ameliorate damage of intestinal structure and immune responses. Overall, these findings are interesting but the following issues need to be addressed.

1. In Fig.1E, the expression of CRAMP was co-localized with DAPI instead of E-cadherin, whether CRAMP could be translocated into nucleus?
2. In Fig.3, despite the production of duodenal CRAMP in CRAMP/gluten group is more than that in CRAMP (prophy)/gluten group, no significant different occurs in their therapeutic effect. So the timing of CRAMP administration is more important for GIE treatment?
3. Could CRAMP treatment lead to the alteration of gut microbiota and how dose gut microbiota regulate the CRAMP production in intestinal epithelium? Metabolites or other mediators? Whether endogenous CRAMP from intestinal epithelium is required for inhibiting *Pseudomonas aeruginosa*, which protects intestinal epithelium and furthermore contributed to intestinal CRAMP production.
4. As shown in all flow cytometry data, the percentage of immune cells was presented, the authors should also provide the absolute number of these cells. Moreover, what are mechanisms by which CRAMP administration modulate macrophages, Tregs and DCs.
5. Can CRAMP be used in clinical practice? Please comment

Referee #2 (Remarks for Author):

The authors showed an interesting and complete study on the role of cathelicidin-related antimicrobial peptide (CRAMP) in gluten-induced enteropathy (GIE). The study demonstrated that CRAMP production was defective in GIE and CRAMP administration ameliorated GIE. The authors further provided evidence that GIE-associated gut dysbiosis contributed to defective intestinal CRAMP production and GIE development. Thus, gut microbiota-CRAMP axis represents a potential therapeutic strategy for human GIE (celiac disease). The experimental designs are logical and well-described. In general, the data are convincing and support their conclusion. However, a few concerns need to be addressed.

1. In Figure 1, epithelial CRAMP production was defective in mice with GIE (sensitized and maintained on a gluten diet), compared with control mice (sensitized but maintained on a gluten-free diet). What is the level of CRAMP in non-sensitized mice on a gluten vs gluten-free diet? This could clarify if the defect was due to T cell-mediated sensitization (or not).
2. The authors demonstrated that CRAMP inhibited the expression of IL-15 using ex vivo isolated epithelial cells from mouse duodenum. This is an important observation and could this be repeated in human epithelial cells?
3. In Figure 4, LL-37 reduced EGFR phosphorylation and MyD88 expression via MMP activity. The authors further discussed that MyD88 is known to be associated with zonulin. It will be more convincing if the authors can provide direct evidence that LL-37 regulates zonulin in human epithelial cells.
4. Was the exogenous CRAMP peptide pro-form or mature form? The authors should add more precise information on the CRAMP peptide used in this study.
5. Please specify whether the intestinal microbiota in Figure 6 and S5 was extracted from duodenum or feces.
6. The microbiota sequencing study showed mice with GIE had increased *Pseudomonas aeruginosa* and reduced *Akkamansia muciniphila*. *P. aeruginosa* was shown to modulate CRAMP. Do *A. muciniphila* play any role in regulating CRAMP expression, GIE, or modulation of *P. aeruginosa*? This should be discussed.

***** Reviewer's comments *****

Referee #1 (Comments on Novelty/Model System for Author):

This manuscript by Ren and colleagues investigates the function of gut microbiota-CRAMP-mediated modulation of intestinal barrier function and immune responses during gluten-induced enteropathy. The authors find that mice with GIE have altered microbiota compositions, which contributes to the CRAMP degradation in intestinal epithelium. Most intriguingly, they report that exogenous CRAMP treatment markedly ameliorates damage of intestinal structure and immune responses. Overall, these findings are interesting but the following issues need to be addressed.

Thank you very much for taking time to review this article and for your valuable comments.

1. In Fig.1E, the expression of CRAMP was co-localized with DAPI instead of E-cadherin, whether CRAMP could be translocated into nucleus?

Ans: Thank you. We and other research groups have reported that although CRAMP can form complexes with released DNA in diseases such as psoriasis and systemic lupus erythematosus (PMID: 17873860; 21389263), it is found to be present in the cytoplasmic compartment in many other conditions (PMID: 22927244; 28069814; 26253786; 31976546; 31420464). As rightly pointed out by you, localization of CRAMP was not well represented by earlier graphs due to the use of slice scanner. Time taken for drying of slides as required by the slice scanner may have caused fluorescence quenching and artifacts. We have repeated the immunofluorescent experiments on CRAMP localization using an ultra-high-resolution confocal microscope (LSM880, Carl Zeiss, Germany) to timely capture the staining. As shown in the revised Figure 1E, CRAMP was co-localized primarily with E-cadherin. The related information has also been updated in the revised expanded view files.

Figure 1E. Localization and expression of CRAMP (red), E-cadherin (green), F4/80 (green) and Ly6G (green) in duodenum by immunofluorescent staining. Representative photomicrographs of individual and merged staining were shown. Nuclei were stained with DAPI (blue). Scale bar: 50 μ m.

2. In Fig.3, despite the production of duodenal CRAMP in CRAMP/gluten group is more than that in CRAMP (prophy)/gluten group, no significant different occurs in their therapeutic effect. So, the timing of CRAMP administration is more important for GIE treatment?

Ans: Thank you. In our study design, CRAMP/gluten group and CRAMP (prophy)/gluten group received 7 injections of CRAMP (0-6 weeks, once a week) after adoptive T cells transfer and 2 injections of CRAMP (-2 and -1 weeks, once a week) before adoptive T cells transfer, respectively (Figure 3A). Not surprisingly, CRAMP as determined at the end of treatment (6 weeks after adaptive gliadin-specific T cells transfer) was higher in CRAMP/gluten group than in CRAMP (prophy)/gluten group (Figure 3B). Comparing these two groups, we observed more pronounced effect of CRAMP given as in CRAMP/gluten group in the upregulation of duodenal tight junction proteins occludin and ZO-2 than in CRAMP (prophy)/gluten group (Figure 3E). For other markers, comparable effects were observed between the two treatment groups. We agree with you that the timing of CRAMP administration is important for GIE treatment.

Figure 3A. Animal protocol

3. Could CRAMP treatment lead to the alteration of gut microbiota and how dose gut microbiota regulates the CRAMP production in intestinal epithelium? Metabolites or other mediators? Whether endogenous CRAMP from intestinal epithelium is required for inhibiting *Pseudomonas aeruginosa*, which protects intestinal epithelium and furthermore contributed to intestinal CRAMP production.

Ans: CRAMP, an endogenous antimicrobial peptide, has been shown to have an important role in the maintenance of gut microbiota homeostasis (PMID: 29440355; 33292444; 15814717; 31679249). CRAMP treatment could lead to the alteration of gut microbiota. For example, CRAMP administration attenuated enterohemorrhagic *Escherichia coli*-induced microbiota disruption (PMID: 28062699) and inhibited colitis-associated microbiota increase (PMID: 22507188).

How microbiota regulates intestinal CRAMP production is still not clearly demonstrated. However, indirect *in vitro* evidence has shown that gut pathogens may produce proteases to degrade and inactivate cathelicidin, such as *Pseudomonas aeruginosa*-stimulated LasB, *Proteus mirabilis*-stimulated ZapA and *Streptococcus pyogenes*-stimulated SpeB

(PMID: 19756242; 12366839). Here we demonstrated that feeding mice with two *P. aeruginosa* strains caused reduced duodenal production of CRAMP (Figure 6I) and the degradation of CRAMP in GIE could be attributable to the production of the protease LasB by *P. aeruginosa* (Figure 6E).

An important role of endogenous CRAMP against *P. aeruginosa* has been demonstrated in other disease contexts, such as lung infection and keratitis (PMID: 22634613; 17898271). Our ongoing unpublished data have suggested that CRAMP deficient mice had increased *P. aeruginosa*, and thus a role for endogenous CRAMP in inhibiting *P. aeruginosa*. Although CRAMP and *P. aeruginosa* are likely mutually regulating, we would prefer to keep the focus of this manuscript on the role of gut microbiota-CRAMP axis on GIE. Effect of CRAMP deficiency or exogenous CRAMP treatment on *P. aeruginosa* during GIE will be thoroughly reported as a follow-up study.

4. As shown in all flow cytometry data, the percentage of immune cells was presented, the authors should also provide the absolute number of these cells. Moreover, what are mechanisms by which CRAMP administration modulate macrophages, Tregs and DCs.

Ans: Thank you. We used the dot plot to present flow cytometry data. The total cell number in one FACS figure was constantly 10,000 (10^4), and the absolute cell number was percentage x 10^4 .

Our data (Figure 5C and 5D) suggested that CRAMP promoted the modulatory phenotypes of aldehyde dehydrogenase (ALDH)⁺ macrophages and DCs. ALDH is a key enzyme for retinoic acid production, which has an important role in inducing Treg. Also, our data suggested that CRAMP promoted conversion of cDCs to the modulatory phenotype (CD103⁺ DCs) (Figure 5C), which is important for retinoic acid-dependent Treg generation (PMID: 17620362; 20068222). Earlier studies have suggested additional mechanisms of cathelicidin by regulating the TLR4 ligand LPS activity (PMID: 21441450) or by regulating the translocation of NF- κ B subunit (PMID: 16456005) to modulate the recruitment and phenotypes of macrophages.

5. Can CRAMP be used in clinical practice? Please comment

Ans: As far as we know, the effect and efficacy of the human ortholog of CRAMP, LL-37 have been evaluated or are under evaluation in clinical trials. The beneficial effect of LL-37 in promoting wound healing has been confirmed in venous leg ulcers (Akademiska Hospital, Sweden; PMID: 25041740). A phase II clinical trial on the therapeutic effect of LL-37 against melanoma was completed in December, 2020 (University of Texas MD Anderson Cancer Center, USA; <https://clinicaltrials.gov/ct2/show/NCT02225366?term=LL-37&draw=2&rank=2>). The effect of LL-37 on bacteria colonization, inflammation response and healing rate of diabetic foot ulcers is currently under evaluation in a phase II clinical trial in Fakultas Kedokteran Universitas (Indonesia; <https://clinicaltrials.gov/ct2/show/NCT04098562?term=LL-37&draw=2&rank=4>).

Referee #2 (Remarks for Author):

The authors showed an interesting and complete study on the role of cathelicidin-related antimicrobial peptide (CRAMP) in gluten-induced enteropathy (GIE). The study demonstrated that CRAMP production was defective in GIE and CRAMP administration ameliorated GIE. The authors further provided evidence that GIE-associated gut dysbiosis contributed to defective intestinal CRAMP production and GIE development. Thus, gut microbiota-CRAMP axis represents a potential therapeutic strategy for human GIE (celiac disease). The experimental designs are logical and well-described. In general, the data are convincing and support their conclusion. However, a few concerns need to be addressed.

Thank you very much for taking time to review this article and for your valuable comments.

1. In Figure 1, epithelial CRAMP production was defective in mice with GIE (sensitized and maintained on a gluten diet), compared with control mice (sensitized but maintained on a gluten-free diet). What is the level of CRAMP in non-sensitized mice on a gluten vs gluten-free diet? This could clarify if the defect was due to T cell-mediated sensitization (or not).

Ans: Thank you for this suggestion. We have performed additional experiments to detect and compare CRAMP levels in unsensitized *Rag1*^{-/-}/gluten-free and unsensitized *Rag1*^{-/-}/gluten mice (Figure EV1A). There was no significant difference between the two groups, suggesting that the defect in CRAMP production was dependent on sensitization by gliadin-specific T cells.

Figure EV1A. Serum CRAMP determination by ELISA. *Rag1*^{-/-}/gluten-free: unsensitized *Rag1*^{-/-} mice fed with gluten-free diet. *Rag1*^{-/-}/gluten: unsensitized *Rag1*^{-/-} mice fed with gluten-containing diet. Data were representative and were the mean \pm SD of three independent experiments with eight mice per group in each experiment. *P* values were calculated using two-tailed *t*-test.

2. The authors demonstrated that CRAMP inhibited the expression of IL-15 using ex vivo isolated epithelial cells from mouse duodenum. This is an important observation and could this be repeated in human epithelial cells?

Ans: Thank you. We have performed the experiment to analyze IL-15 expression in human epithelial cells and the new data have been added in new Figure EV3B. It was observed that LL-37 (the human ortholog of CRAMP) inhibited *IL15* expression in human

epithelial cells.

Figure EV3B. The mRNA levels of *IL15* *in vitro* were measured by RT-qPCR. Data were representative and were the mean \pm SD of three independent experiments. *P* values were calculated using ANOVA test with correction for multiple comparisons.

3. In Figure 4, LL-37 reduced EGFR phosphorylation and MyD88 expression via MMP activity. The authors further discussed that MyD88 is known to be associated with zonulin. It will be more convincing if the authors can provide direct evidence that LL-37 regulates zonulin in human epithelial cells.

Ans: Thank you for this suggestion. We have performed additional experiments on the expression of zonulin by Western blot. As shown in the new Figure 4B, zonulin was downregulated by LL-37 in human epithelial cells.

4. Was the exogenous CRAMP peptide pro-form or mature form? The authors should add more precise information on the CRAMP peptide used in this study.

Ans: We used the mature form of CRAMP for exogenous treatment. We have added this description in 'Materials & Methods' in the 'revised manuscript'.

5. Please specify whether the intestinal microbiota in Figure 6 and S5 was extracted from duodenum or feces.

Ans: The intestinal microbiota was extracted from duodenal content. We have specified this information in 'expanded view materials&methods' in 'revised expanded view files'.

6. The microbiota sequencing study showed mice with GIE had increased *Pseudomonas aeruginosa* and reduced *Akkamansia muciniphila*. *P. aeruginosa* was shown to modulate CRAMP. Do *A. muciniphila* play any role in regulating CRAMP expression, GIE, or modulation of *P. aeruginosa*? This should be discussed.

Ans: Thank you for this comment. Indeed, there have been many studies on the protective effect of *A. muciniphila* on the intestinal barrier function (PMID: 23671105; 29472701; 31632373). However, the effect of *A. muciniphila* on CRAMP, GIE and *P. aeruginosa* is still unknown. In fact, ongoing work of our group suggested that the protective mechanisms of live *A. muciniphila* and pasteurized *A. muciniphila* on GIE were different. To keep a focused goal of this work, we will report the data as a follow-up study.

12th May 2021

Dear Prof. Sun,

Thank you for the submission of your revised manuscript to EMBO Molecular Medicine. I am pleased to inform you that we will be able to accept your manuscript pending the following final amendments:

- 1) Tables: Please rename Expanded View Files to "Appendix" with a table of content and rename tables to "Appendix Table S1 and S2". Also, correct their callouts in the text accordingly.
- 2) In the main manuscript file, please do the following:
 - Correct/answer the track changes suggested by our data editors by working from the attached/uploaded document.
 - Remove font colour.
 - Specify author contributions for He Liu and Hanfei Li i.e., HeLi. and HaLi.
 - Data availability section should include information only about datasets deposited in public repositories. Please use the following format to report the accession number of your data:

The datasets produced in this study are available in the following databases:
[data type]: [full name of the resource] [accession number/identifier] ([doi or URL or identifiers.org/DATABASE:ACCESSION])

Please check "Author Guidelines" for more information.

<https://www.embopress.org/page/journal/17574684/authorguide#availabilityofpublishedmaterial>

- Please merge "Funding" section with "Acknowledgements".
- 3) Source data: Please upload one file per figure.
 - 4) Synopsis: Please check your synopsis text and image, revise them if necessary and submit their final versions with your revised manuscript. Please be aware that in the proof stage minor corrections only are allowed (e.g., typos).
 - Synopsis image: Please consider revising the background colours (no colours is also an option) used in the graphical abstract and resize the image to 550 px-wide x (250-400)-px high.
 - 5) For more information: There is space at the end of each article to list relevant web links for further consultation by our readers. Could you identify some relevant ones and provide such information as well? Some examples are patient associations, relevant databases, OMIM/proteins/genes links, author's websites, etc...
 - 6) As part of the EMBO Publications transparent editorial process initiative (see our Editorial at <http://embomolmed.embopress.org/content/2/9/329>), EMBO Molecular Medicine will publish online a Review Process File (RPF) to accompany accepted manuscripts. This file will be published in conjunction with your paper and will include the anonymous referee reports, your point-by-point response and all pertinent correspondence relating to the manuscript. Let us know whether you agree with the publication of the RPF and as here, if you want to remove or not any figures from it prior to publication. Please note that the Authors checklist will be published at the end of the RPF.
 - 7) Please provide a point-by-point letter INCLUDING my comments as well as the reviewer's reports and your detailed responses (as Word file).

I look forward to reading a new revised version of your manuscript as soon as possible.

Yours sincerely,

Zeljko Durdevic

***** Reviewer's comments *****

Referee #1 (Remarks for Author):

In this revised the manuscript, the authors addressed all my concerns.

Referee #2 (Remarks for Author):

The authors have addressed all my concerns and the quality of the revised manuscript has been greatly improved. I think this manuscript fits well within EMBO Molecular Medicine as the authors focus primarily on studies that provide functional novel insights of translational significance in an appropriate in vivo and in vitro model, which are also conceptually novel and of broad interest. Therefore, I think this manuscript could be considered for publication as a regular paper in EMBO Molecular Medicine.

The authors performed the requested editorial changes.

We are pleased to inform you that your manuscript is accepted for publication and is now being sent to our publisher to be included in the next available issue of EMBO Molecular Medicine.

Corresponding Author Name: Jia Sun, Julien Diana

Manuscript Number: EMM-2021-14059